# Value of Mini Electrodes for Mapping Myocardial Arrhythmogenic Substrate—The Influence of Tip-to-Tissue Angulation and Irrigation Flow on Signal Quality

Karen Bickel [1], Thorsten Lewalter [1], Johannes Fischer [2], Christine Baumgartner [2], Petra Hoppmann [3], Klaus Tiemann [1] and Clemens Jilek [1,*]

[1] Peter-Osypka Heart Center Munich, Internistisches Klinikum München Süd, 81379 München, Germany; karen.bickel@t-online.de (K.B.); thorsten.lewalter@ikms.de (T.L.); klaus.tiemann@ikms.de (K.T.)
[2] Zentrum für Präklinische Forschung, Klinikum rechts der Isar, Technische Universität München, 81675 München, Germany; fischer.johannes@tum.de (J.F.); christine.baumgartner@tum.de (C.B.)
[3] Klinik und Poliklinik für Innere Medizin I, Klinikum rechts der Isar, Technische Universität München, 81675 München, Germany; petra.hoppmann@tum.de
[*] Correspondence: research@jilek.de

**Abstract:** Background: The use of mini electrodes with a small surface and narrow electrode-to-electrode spacing is believed to lead to a higher electrical resolution. Until now, the effects of tip-to-tissue contact, angulation, and irrigation on signal quality and morphology are unknown. Methods: The beating heart of an open-chest pig was examined while controlling the angulation and contact between the catheter tip and myocardial tissue, as well as the irrigation of the catheter tip. The mini electrodes were mounted onto commercially available 8 mm non-irrigated and 4 mm irrigated tip catheters. Different electrode interconnections, angulations, contact forces, and irrigation flow were analyzed and compared to signals recorded from conventional electrodes. Results: A total of 63 electrode samples of 21 defined, stable settings, each lasting 30 s, were analyzed. (1) Tissue contact of mini electrodes was given as soon as the conventional tip electrode showed tissue contact. (2) Angulation of the tip-to-tissue contact showed a trend towards changes in the integral of signals derived from mini electrodes, and no significant changes were seen in signals derived from conventional or mini electrodes. (3) Irrigation flow surrounding the mini electrodes did not influence signals derived from mini electrodes, whereas conventional electrodes showed signals with a longer duration under higher irrigation. Conclusion: Mini electrodes are robust to contact force and irrigation flow regarding signal quality, whereas signals of conventional electrodes are affected by irrigation flow, leading to substantial changes in signal duration and kurtosis. Signals of mini electrodes are sensitive to local electrical changes because of a high local resolution.

**Keywords:** mini electrode; signal morphology; irrigation; angulation; contact force

## 1. Introduction

Catheter-based ablation is the standard treatment for many cardiac arrhythmias [1,2]. Intracardiac electrograms derived from electrodes at catheters represent a cornerstone in electrophysiological (EP) and ablation procedures to identify arrhythmogenic substrate and control the success of ablation. Standard EP catheters are equipped with ring electrodes and, in the case of ablation, an additional tip electrode with a length ranging from 3.5 mm up to 10 mm. The electrodes are usually interconnected in a bipolar manner, meaning along the longitudinal axis of the catheter.

Mini electrodes, showing a small surface area, are used for high-density mapping and are usually not configured as ring electrodes. In the case of ablation catheters, the mini electrodes are mounted at the tip electrode as three circumferentially placed circles with an electrode diameter of 1 mm. Mini electrodes are believed to lead to a higher electrical resolution because of the smaller electrode surface and the more narrow electrode spacing

compared with conventional electrode designs, and they are further supposed to reduce far field signals [3]. Mini electrodes have been shown to increase sensitivity in the detection of scars and in the identification of small near-field components that are missed with conventional large tip electrodes in a post-infarction pig model [4,5].

In animal models, changes in the intracardiac electrograms derived from mini electrodes could be taken as a surrogate parameter for monitoring of ablation lesion maturation [6,7] and may prevent unnecessarily prolonged ablation application [5], thus reducing extracardiac injuries [6].

In human ablation procedures, mini electrode technology mounted on the tip electrode has displayed a significantly shorter radiofrequency time and time to achieve cavotricuspid isthmus block when attempting typical cavotricuspid-isthmus (CTI)-dependent atrial macro-reentry tachycardia (MRT) in a non-randomized, cohort study [8]. The findings of Takagi agree with the theoretical perception that the electrical current of CTI-dependent MRT has the highest amplitude in the interconnections of mini electrodes because the vector of the electrical wavefront of CTI-dependent MRT is longitudinal to the mini electrodes, whereas it is perpendicular to the conventional electrodes.

In the setting of AV node re-entrant tachycardia ablation, mini electrodes could identify a higher proportion of slow pathway potentials compared with conventional bipolar mapping, and could also identify double potentials, which may mark a clinical endpoint [9].

Until now, some important basic questions about electrical signals derived from mini electrodes have not been investigated. In our animal work, we aimed to clarify the following:

- The impact of the contact of mini electrodes on signal quality, because whether mini-electrode-to-tissue-contact is ensured when conventional electrodes show tissue contact is not tested;
- The influence of the angulation of the catheter tip on the signal morphology of mini electrodes;
- The influence of irrigation on the signal morphology of mini electrodes, as the electrodes are widely surrounded by the irrigation fluid;
- The signal interconnection that ensures a stable and reliable signal.

## 2. Materials and Methods

### 2.1. Animal Preparation and Setting

The animal trial was approved by the Government of Upper Bavaria (approval number ROB-55.2-2532. Vet_02-1 7-174).

A beating heart of an open-chest pig was examined, mainly epicardially, controlling the angulation and contact between the catheter tip and myocardial tissue, as well as the irrigation of the catheter tip. In a pre-test, it was assured that there were no significant differences between signals derived from mini electrodes from the endocardium and epicardium.

The pig was pre-treated with an intramuscular injection of 10–15 mg/kg ketamine, 2 mg/kg azaperone, and 1 mg atropine. The examination was carried out under general anesthesia maintained with propofole (12.5 mg/kg/h) and controlled ventilation. Analgesia was ensured with metamizole (40 mg/kg) and fentanyle (0.015 mg/kg) administered around every 30 min, with the addition of an intravenous fluid substitution of 10 mL/kg/h.

A lateral, partial thoracotomy was performed, and access to the epicardial anterior left atrium was gained. Sutures were set at the anterior wall of the left atrium to ensure fixation of the epicardial myocardium of the left atrium to ensure a stable mapping position of the catheter. The heart had to be in sinus rhythm throughout the whole procedure.

At the end of the procedure, the pig was sacrificed by phentobarbitale i.v. (0.1 mL/kg) followed by 40 mL of 7.45% potassium chloride intravenously.

### 2.2. Catheter Examination

Catheters with mounted mini electrodes at a non-irrigated 8 mm tip (Intellatip MiFi XP, Boston Scientific, Marlborough, MA, USA; see Figure 1) and at a irrigated 4 mm tip (Intellatip MiFi OI, Boston Scientific, Marlborough, MA, USA) were investigated. An adjustable flow pump (HAT 500 System, Osypka AG, Rheinfelden, Germany) was used for irrigation. Signal processing and management was performed with the EP system LS Pro and the signal amplifier STAMP (former C.R. Bard, now Boston Scientific, Marlborough, MA, USA).

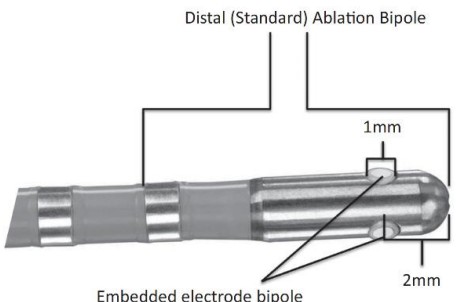

**Figure 1.** Mini electrodes mounted at a 8 mm tip of an ablation catheter (Intellatip MiFi XP, Boston Scientific, Marlborough, MA, USA).

The protocol consisted of several steps, and each condition of every step was performed three times under stable catheter settings for at least 30 s.

### 2.3. Effect of Contact Force on Signals of Mini Electrodes

The hypothesis was that, with low contact force, there could be contact of the conventional tip electrode, but loss of contact of the mini electrodes. The tip electrode was placed on the epicardial myocardium (1) with a force such that only the tip electrode, not the mini electrode, made contact with the tissue and (2) with a force such that both the conventional tip and mini electrodes made contact with the tissue.

### 2.4. Effect of Catheter Angulation on Mini Electrodes

Three angulations with a tangential, 45°, and 90° position were tested with regard to the duration, amplitude, integral and kurtosis of signals derived from mini electrodes and conventional electrodes. Both catheters, with mounted electrodes at a 8 mm non-irrigated tip and a 4 mm irrigated tip, were used (see Figure 2).

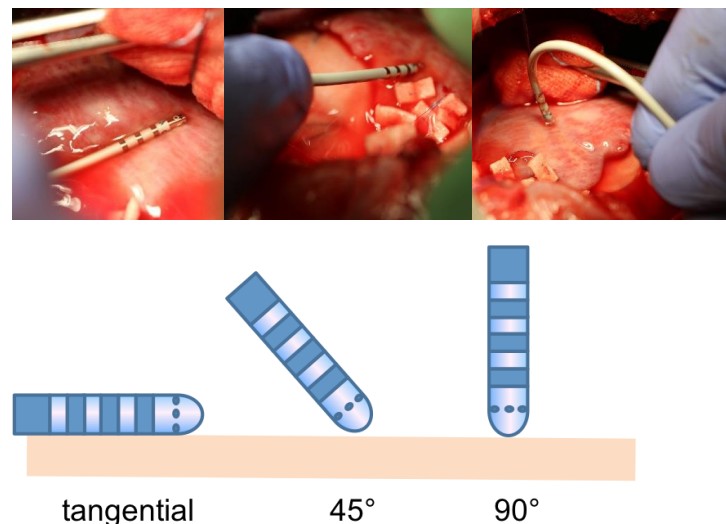

**Figure 2.** Setting of three angulations of the tip catheter to epicardial tissue.

### 2.5. Effect of Irrigation Flow on Mini Electrodes

The 4 mm tip irrigated catheter was evaluated with regard to the respective effects on signal parameters under flow rates of 2 mL/min and 10 mL/min.

### 2.6. Effect of Electrode Interconnections on Signal Quality and Stability

The hypothesis was that the variation in signal parameters is higher among signals derived from mini electrodes compared with those from conventional electrodes.

### 2.7. Signal Filtering and Processing and Statistics

Bipolar signals were recorded with a recording sensitivity of 5 mV for both unipolar and bipolar signals. The low cutoff was 1 Hz for unipolar and 40 Hz for bipolar signals, while the high cutoff was 25 Hz and 250 Hz, respectively. A notch filter was not used.

For every condition, a segment with a stable zero line and artifact-free signals was chosen with a length of 4 s, exported, and analyzed with Mathlab (The MathWorks, Inc., Ismaning, Germany). The maximum, minimum, integral of positive and negative, and sum of the integral above the zero line and the absolute value of the integral below the zero line, as well as the signal duration and the kurtosis of the signal, were calculated for every beat. The amplitudes are presented in mV, duration in msec, and integral in $mV^2 \cdot msec$.

The coefficient of variation was calculated for the duration, amplitude, and kurtosis as a parameter of signal stability.

All variables were calculated as means and standard deviation, as median when indicated and in the case of a non-normal distribution. Normal distribution was tested by the Kolmogorow–Smirnow test. Confidence intervals of 95% were calculated. For multiple group comparisons of normally distributed data, repeated measures ANOVA was used. In the case of significant findings, paired group comparisons with Student's *t*-test were performed. A *p*-value $\leq 0.05$ was considered significant.

### 3. Results

In total, 63 electrode samples of 21 defined, stable settings, each lasting 30 s, were analyzed. Figure 3 shows an example of the signals derived from mini and conventional electrodes, showing that the ventricular far field signal is more pronounced in signals derived from conventional electrodes compared with those from mini electrodes. The results are visualized in a central figure (see Figure 3), and the exact data are shown in the corresponding paragraph.

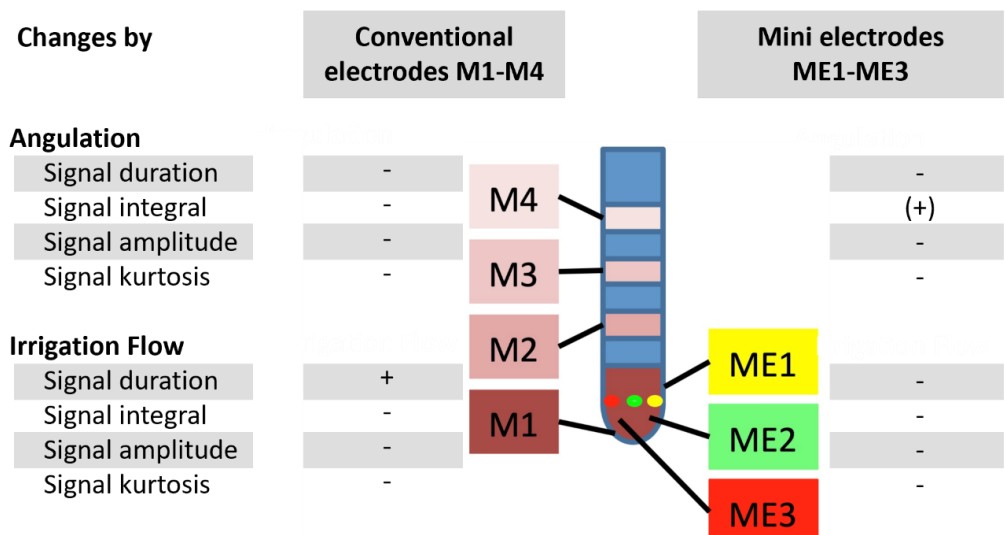

**Figure 3.** Overview of signal changes in conventional and mini electrodes by angulation and irrigation flow. Abbreviations: $-$ = no change, + = significant change, (+) = trend to a change.

### 3.1. Effect of Contact Force on Signals of Mini Electrodes

Efforts were taken to ensure contact and a stable signal with the conventional tip electrode with no contact and no signal on the mini electrodes. Despite several maneuvers and angulation while fixating a left atrial wall segment, it was not possible to realize a stable situation in this beating heart model. Whenever the conventional tip electrode displayed tissue contact and a stable signal, the mini electrodes also displayed tissue contact and a stable signal.

### 3.2. Effect of Tip Angulations on Mini Electrodes

Three angulations (tangential, 45°, and 90°) were compared using the 8 mm non-irrigated Intellatip MiFi XP tip and the 4 mm irrigated Intellatip MiFi OI tip catheter.

Overall, no significant effect of angulation could be observed on conventional and mini electrodes using the 8 mm or 4 mm tip electrode catheter. In ANOVA, the mini electrodes mounted on an 8 mm tip electrode showed a trend to higher integrals in the 90° position compared with the tangential position.

The signal parameters are listed in Tables 1–4.

**Table 1.** Signal Duration: Comparison of IntellaTip MiFi XP (8 mm tip) and Intellatip MiFi OI (4 mm tip, 2 ml/min flow rate) with different angulations (tangential, 45°, and 90°): mean and standard deviation (SD) of electrograms from conventional bipolar tip electrodes (MAP 1/2) as well as from mini electrodes (ME 1/2, ME 2/3, ME 3/1). Statistics: repeated ANOVA, and in the case of significance, paired Student's $t$-test, $p \leq 0.05$ considered significant.

| Channel | Tangential | | 45 Degree | | 90 Degree | | $p$ ANOVA |
|---|---|---|---|---|---|---|---|
| | Duration Mean | Duration SD | Duration Mean | Duration SD | Duration Mean | Duration SD | |
| **MiFi XP** | | | | | | | |
| MAP 1/2 | 23.47 | 24.62 | 34.27 | 32.45 | 71.00 | 43.59 | 0.278 |
| ME 1/2 | 75.20 | 73.69 | 14.27 | 2.96 | 15.13 | 3.20 | 0.214 |
| ME 2/3 | 59.87 | 55.93 | 24.07 | 25.29 | 16.67 | 2.89 | 0.353 |
| ME 3/1 | 89.00 | 67.34 | 25.60 | 24.80 | 21.80 | 2.94 | 0.159 |
| **MiFi OI** | | | | | | | |
| MAP 1/2 | 33.73 | 25.14 | 21.13 | 3.96 | 20.47 | 2.20 | 0.524 |
| ME 1/2 | 70.80 | 76.54 | 15.87 | 5.46 | 26.53 | 18.61 | 0.359 |
| ME 2/3 | 68.87 | 67.70 | 43.40 | 39.40 | 23.80 | 15.01 | 0.525 |
| ME 3/1 | 63.07 | 55.31 | 28.20 | 25.07 | 22.40 | 15.42 | 0.385 |

**Table 2.** Signal Integral: Comparison of IntellaTip MiFi XP (8 mm tip) and Intellatip MiFi OI (4 mm tip, 2 ml/min flow rate) with different angulations (tangential, 45°, and 90°): mean and standard deviation (SD) of electrograms from conventional bipolar tip electrodes (MAP 1/2) as well as from mini electrodes (ME 1/2, ME 2/3, ME 3/1). Statistics: repeated ANOVA, and in the case of significance, paired Student's $t$-test, $p \leq 0.05$ considered significant.

| Channel | Tangential | | 45 Degree | | 90 Degree | | $p$ tang vs. 45 | $p$ tang vs. 90 | $p$ 45 vs. 90 | ANOVA |
|---|---|---|---|---|---|---|---|---|---|---|
| | Integral Mean | Integral SD | Integral Mean | Integral SD | Integral Mean | Integral SD | | | | |
| **MiFi XP** | | | | | | | | | | |
| MAP 1/2 | 32.97 | 20.54 | 16.68 | 10.63 | 8.60 | 2.63 | | | | 0.165 |
| ME 1/2 | 2.34 | 2.64 | 7.38 | 5.32 | 12.15 | 4.44 | 0.216 | 0.030 | 0.299 | 0.071 |
| ME 2/3 | 3.81 | 6.22 | 10.52 | 8.31 | 5.68 | 4.53 | | | | 0.453 |
| ME 3/1 | 1.12 | 1.60 | 3.85 | 2.83 | 3.77 | 1.78 | | | | 0.471 |

Table 2. *Cont.*

| Channel | Tangential | | 45 Degree | | 90 Degree | | *p* tang vs. 45 | *p* tang vs. 90 | *p* 45 vs. 90 | ANOVA |
|---|---|---|---|---|---|---|---|---|---|---|
| | Integral Mean | Integral SD | Integral Mean | Integral SD | Integral Mean | Integral SD | | | | |
| MiFi OI MAP 1/2 | 28.29 | 22.63 | 32.44 | 17.57 | 25.51 | 11.18 | | | | 0.892 |
| ME 1/2 | 4.86 | 4.69 | 3.48 | 2.71 | 2.26 | 0.98 | | | | 0.629 |
| ME 2/3 | 6.72 | 6.48 | 2.63 | 0.52 | 4.21 | 2.55 | | | | 0.497 |
| ME 3/1 | 7.66 | 9.70 | 4.27 | 2.67 | 5.24 | 1.92 | | | | 0.824 |

**Table 3.** Signal Kurtosis: Comparison of IntellaTip MiFi XP (8 mm tip) and Intellatip MiFi OI (4 mm tip, 2 ml/min flow rate) with different angulations (tangential, 45°, and 90°): mean and standard deviation (SD) of electrograms from conventional bipolar tip electrodes (MAP 1/2) as well as from mini electrodes (ME 1/2, ME 2/3, ME 3/1). Statistics: repeated ANOVA, and in the case of significance, paired Student's *t*-test, $p \leq 0.05$ considered significant.

| Channel | Tangential | | 45 Degree | | 90 Degree | | ANOVA |
|---|---|---|---|---|---|---|---|
| | Kurtosis Mean | Kurtosis SD | Kurtosis Mean | Kurtosis SD | Kurtosis Mean | Kurtosis SD | |
| MiFi XP | | | | | | | |
| MAP 1/2 | 22.72 | 2.58 | 28.12 | 10.40 | 25.47 | 7.35 | 0.640 |
| ME 1/2 | 33.02 | 19.57 | 36.43 | 16.30 | 31.33 | 8.65 | 0.926 |
| ME 2/3 | 39.76 | 25.71 | 43.96 | 16.70 | 30.13 | 6.48 | 0.684 |
| ME 3/1 | 30.78 | 19.04 | 35.44 | 14.03 | 22.99 | 9.70 | 0.233 |
| MiFi OI | | | | | | | |
| MAP 1/2 | 22.08 | 3.36 | 20.05 | 4.09 | 18.03 | 2.67 | 0.404 |
| ME 1/2 | 19.19 | 9.07 | 33.14 | 11.85 | 21.46 | 7.19 | 0.231 |
| ME 2/3 | 28.92 | 24.10 | 42.15 | 24.63 | 24.27 | 4.35 | 0.548 |
| ME 3/1 | 42.22 | 20.44 | 30.06 | 10.08 | 43.89 | 10.59 | 0.513 |

**Table 4.** Signal Amplitude: Comparison of IntellaTip MiFi XP (8 mm tip) and Intellatip MiFi OI (4 mm tip, 2 ml/min flow rate) with different angulations (tangential, 45°, and 90°): mean and standard deviation (SD) of electrograms from conventional bipolar tip electrodes (MAP 1/2) as well as from mini electrodes (ME 1/2, ME 2/3, ME 3/1). Statistics: repeated ANOVA, and in the case of significance, paired Student's *t*-test, $p \leq 0.05$ considered significant.

| Channel | Tangential | | 45 Degree | | 90 Degree | | ANOVA |
|---|---|---|---|---|---|---|---|
| | Amplitude Mean | Amplitude SD | Amplitude Mean | Amplitude SD | Amplitude Mean | Amplitude SD | |
| MiFi XP | | | | | | | |
| MAP 1/2 | 4.32 | 1.89 | 3.03 | 1.06 | 2.24 | 0.30 | 0.202 |
| ME 1/2 | 1.08 | 0.90 | 2.05 | 0.83 | 2.65 | 0.54 | 0.108 |
| ME 2/3 | 1.29 | 1.27 | 2.74 | 1.38 | 1.90 | 0.80 | 0.359 |
| ME 3/1 | 0.81 | 0.64 | 1.64 | 0.79 | 1.40 | 0.46 | 0.438 |
| MiFi OI | | | | | | | |
| MAP 1/2 | 3.51 | 2.05 | 3.81 | 1.24 | 3.56 | 0.89 | 0.962 |
| ME 1/2 | 1.46 | 1.17 | 1.38 | 0.51 | 1.14 | 0.38 | 0.890 |
| ME 2/3 | 1.84 | 1.72 | 1.36 | 0.30 | 1.52 | 0.63 | 0.851 |
| ME 3/1 | 2.01 | 1.95 | 1.56 | 0.63 | 1.79 | 0.44 | 0.881 |

*3.3. Effect of Irrigation Flow on Mini Electrodes*

Irrigation flow rates of 2 mL/min and 10 mL/min surrounding the mini electrodes did not influence the amplitude, integral, kurtosis, or duration of signals derived from mini

electrodes, whereas conventional electrodes showed a longer signal duration with higher irrigation (signal duration 2 ml flow $25 \pm 16$ msec vs. 10 mL flow $56 \pm 54$ msec; $p = 0.000$; see Table 5).

**Table 5.** Comparison of IntellaTip MiFi OI with different irrigation flow rates of 2 ml/min and 10 mL/min: mean and standard deviation (SD) of electrograms from conventional bipolar electrodes (MAP 1/2, MAP 3/4, MAP 2/3) as well as from mini electrodes (ME 1/2, ME 2/3, ME 3/1). Statistics: paired Student's *t*-test, $p \leq 0.05$ considered significant.

| Irrigation | 2 mL Flow | | 10 mL Flow | | | 2 mL Flow | | 10 mL Flow | | |
|---|---|---|---|---|---|---|---|---|---|---|
| Channel | Duration Mean | Duration SD | Duration Mean | Duration SD | *p*-Value | Integral Mean | Integral SD | Integral Mean | Integral SD | *p*-Value |
| MAP 1/2 | 25.11 | 15.68 | 56.04 | 54.54 | 0.000 | 28.75 | 17.58 | 20.41 | 19.89 | 0.037 |
| MAP 3/4 | 25.67 | 31.47 | 34.78 | 41.82 | 0.244 | 11.38 | 10.63 | 11.47 | 17.04 | 0.977 |
| MAP 2/3 | 65.78 | 53.25 | 78.52 | 58.60 | 0.281 | 3.63 | 3.85 | 3.58 | 4.65 | 0.961 |
| ME 1/2 | 37.73 | 50.62 | 51.80 | 55.24 | 0.209 | 3.53 | 3.29 | 3.09 | 3.46 | 0.538 |
| ME 2/3 | 45.36 | 48.70 | 52.33 | 52.44 | 0.513 | 4.52 | 4.30 | 3.20 | 4.02 | 0.135 |
| ME 3/1 | 37.89 | 39.73 | 49.65 | 55.38 | 0.248 | 5.72 | 5.96 | 5.56 | 6.62 | 0.905 |
| MAP 1/2 | 20.06 | 3.74 | 18.62 | 6.25 | 0.188 | 3.65 | 1.46 | 3.02 | 1.62 | 0.058 |
| MAP 3/4 | 34.32 | 11.55 | 24.38 | 10.37 | 0.000 | 2.67 | 1.60 | 2.19 | 1.88 | 0.196 |
| MAP 2/3 | 19.83 | 6.11 | 18.22 | 6.11 | 0.214 | 1.04 | 0.58 | 1.26 | 1.28 | 0.296 |
| ME 1/2 | 24.60 | 11.20 | 26.74 | 13.34 | 0.410 | 1.32 | 0.76 | 1.23 | 0.86 | 0.600 |
| ME 2/3 | 31.78 | 21.04 | 26.87 | 13.96 | 0.192 | 1.57 | 1.07 | 1.18 | 0.77 | 0.045 |
| ME 3/1 | 38.72 | 15.49 | 27.29 | 12.83 | 0.000 | 1.79 | 1.20 | 1.61 | 1.12 | 0.475 |

### 3.4. Effect of Electrode Interconnection on Signal Quality and Stability

The coefficient of variation of bipolar conventional electrode interconnections and interconnections between mini electrodes and the conventional tip electrode compared with bipolar mini electrode interconnections was numerically lower, but not statistically significant with regard to signal amplitude, duration, and kurtosis (see Table 6).

**Table 6.** Comparison of mean coefficient of variation (CV) of amplitude, kurtosis, and duration of signals between interconnection and the corresponding mini electrode bipolar (if no corresponding ME 1/2 was chosen) of a 4 mm irrigated tip with a flow rate of 2 mL/min. Abbreviations: M conventional tip/ring electrodes, ME mini electrodes. *p*-value calculated with paired Student's *t*-test.

| Channel | Amplitude Mean CV | *p*-Value | Kurtosis Mean CV | *p*-Value | Duration Mean CV | *p*-Value |
|---|---|---|---|---|---|---|
| M 1/2 | 0.39 | 0.600 | 0.17 | 0.287 | 0.34 | 0.444 |
| ME 1/2 | 0.53 | n.a. | 0.40 | n.a. | 0.59 | n.a. |
| ME 2/3 | 0.57 | n.a. | 0.59 | n.a. | 0.55 | n.a. |
| ME 3/1 | 0.58 | n.a. | 0.36 | n.a. | 0.77 | n.a. |
| M1-ME1 | 0.25 | 0.493 | 0.37 | 0.908 | 0.31 | 0.156 |
| M1-ME2 | 0.34 | 0.781 | 0.45 | 0.641 | 0.29 | 0.122 |
| M1-ME3 | 0.27 | 0.741 | 0.36 | 0.987 | 0.30 | 0.150 |

## 4. Discussion

To the best of our knowledge, this paper is the first investigating the effects of contact force, tip–tissue angulation, irrigation, and electrode interconnections on the signal morphology of conventional and mini electrodes in a controlled animal model.

The main findings of the work are as follows.

Stable tip-to-tissue contact of mini electrodes is ensured once the tip electrode has stable tissue contact. It was possible to evoke a signal at the conventional tip electrode, but not at the mini electrodes, in healthy tissue.

Our work confirms the hypothesis that mini electrodes are more sensitive to changes in tip–tissue angulation with regard to signal duration and integral compared with conventional electrodes. The angulation does not affect the sharpness of the signal derived from mini electrodes, as proven by an unaffected signal kurtosis. Using the signal analysis of circumferential located mini electrodes might help to define the tip–tissue angulation of the catheter.

Mini electrodes are more robust to irrigation compared with conventional electrodes. None of the signal parameters of duration, integral, amplitude, or kurtosis changed significantly with a higher irrigation flow rate. In contrast, conventional electrodes showed a significantly longer signal duration with a higher irrigation flow rate. The effect is supposed to be even higher when higher irrigation flow rates of 17 mL/min or 30 mL/min are used for ablation. For technical reasons, an irrigation flow of only up to 10 mL/min could be used during the experiment without ablation. Therefore, one may conclude that mini electrodes seem to be more appropriate for signal analysis during ablation than conventional electrodes. The reason for a more stable signal derived from mini electrodes than from conventional electrodes may be the result of the smaller field covered by the mini electrodes. The smaller the detection field, the more chance of having a stable irrigation milieu around the electrodes. As conventional electrodes are a bipole of the tip and the distal ring electrode, but the irrigation fluid covers only the tip electrode, there might be a disturbance directing the ring electrode from which you expect a mixture between blood and irrigation fluid, resulting in a less stable electrical field. Our results are the first in the literature describing the effects of irrigation on signals with conventional electrodes.

Mini electrodes interconnected in a true bipolar manner are able to better reflect more subtle changes in signal morphology than conventional electrodes, as they cover a smaller area. If a more global view is necessary, mini electrodes may be interconnected with the conventional tip to reduce variations. Therefore, mini electrodes in combination with conventional electrodes enhance the electrical resolution and may have the potential to identify critical ablation sites in an easier and more reliable manner. Our findings are in line with computer models [10]. The impact of this finding has to be proven in a further trial with a clinical end point.

Besides the differences described between mini electrodes and conventional electrodes, it is very important to keep in mind that the bipolar vector of mini electrodes mounted on the catheter tip are perpendicular to the bipolar vector of conventional electrodes. Especially, electrical propagation being perpendicular to the catheter as CTI-dependent atrial flutter is best suitable to be analyzed with mini electrodes, as this allows to define the exact location of the catheter with regard to the border zone from CTI to the ventricle and to the inferior vena cava [11].

A second advantage of mini electrodes is the reduction in far field signal by the smaller area of the mini electrode itself and the interelectrode spacing (see Figure 4). Therefore, mini electrodes reflect true local signals more accurately, whereas conventional electrodes cover a large area and produce larger far field signals than mini electrodes [8].

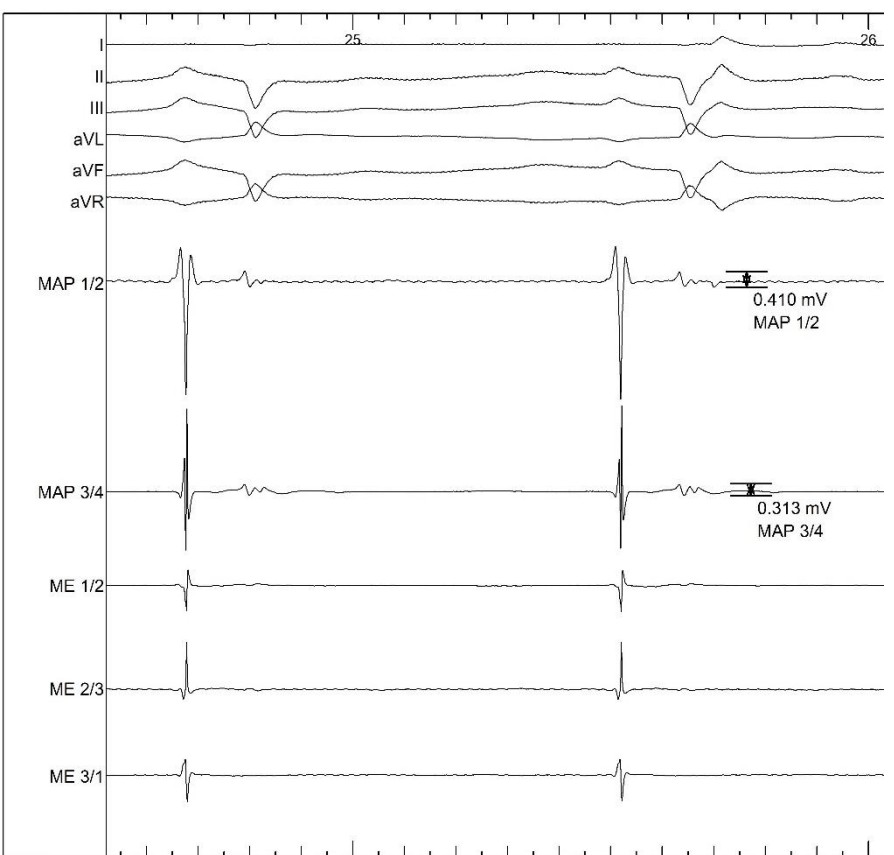

**Figure 4.** Atrial signal derived with more ventricular far field signals on conventional electrodes (MAP 1/2, MAP 3/4) compared with mini electrodes (ME1/2, ME2/3, and ME3/1) from an irrigated 4 mm tip catheter with 2 mL/min flow. Paper speed: 100 mm/s.

*Limitation*

The animal work was performed at the epicardial heart side, making sure to control the tip–tissue contact visually, as both catheters do not allow an intern contact force measurement. An endocardial approach may have effects on signal, whereas our pre-test with a comparison of endocardial and epicardial signals derived from conventional and mini electrodes revealed no significant differences.

For all analyses, mapping of healthy myocardium was performed. This formally restricts the findings to healthy tissue, but is the basis for future research on the signal morphology of altered myocardial tissue such as fibrosis or tissue that has been electrically inactivated by ablation.

Measurements were limited to one pig owing to animal welfare restrictions.

## 5. Conclusions

Mini electrodes are robust to contact force and irrigation flow regarding signal quality, whereas signals of conventional electrodes are affected by irrigation flow, leading to substantial changes in signal duration and kurtosis. Signals of mini electrodes are able to show the angulation of the catheter and are very sensitive to local electrical changes due to a high local resolution.

**Author Contributions:** Conceptualization, K.B., T.L., J.F., C.B., K.T. and C.J.; Data curation, K.B., T.L., K.T. and C.J.; Formal analysis, K.B., T.L., P.H., K.T. and C.J.; Funding acquisition, C.J.; Investigation, K.B., T.L., J.F., C.B., K.T. and C.J.; Methodology, K.B., T.L., J.F., K.T. and C.J.; Project administration, T.L., J.F., C.B., K.T. and C.J.; Resources, T.L and C.J.; Supervision, T.L. and C.J.; Validation, K.B., T.L., P.H. and K.T.; Visualization, K.B.; Writing—original draft, K.B., P.H. and C.J.; Writing—review and

editing, T.L., J.F., C.B., K.T. and C.J. All authors have read and agreed to the published version of the manuscript.

**Funding:** The animal work was funded by Boston Scientific (Marlborough, MA, USA). Grant number #1782252.

**Institutional Review Board Statement:** The animal trial was approved by the Government of Upper Bavaria (approval number ROB-55.2-2532. Vet_02-1 7-174).

**Informed Consent Statement:** Not applicable as study did not involve humans.

**Acknowledgments:** Bernd Bruesehaber (Biotronik, Berlin, Germany) for statistical advice.

**Conflicts of Interest:** T.L. and C.J. received lecture honoraria, educational grants, and/or research grants from Boston Scientific, Abbott, Biotronik, and Medtronic. P.H. received research support from Boston Scientific. All other authors have no conflicts of interest.

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
