# Peer review of "Value of Mini Electrodes for Mapping Myocardial Arrhythmogenic Substrate—The Influence of Tip-to-Tissue Angulation and Irrigation Flow on Signal Quality"

_2813-2475, doi:10.3390/jvd1010002_

Round 1

Reviewer 1 Report

This study reports on tip-to-tissue contact, angulation, and irrigation effects on recorded signal quality and morphology in cardiac mapping. 

This work is well designed and executed study that would advance the reproducibility and vigor of cardiac mapping studies. However, authors should improve statistical analysis and its description to meet accepted publication standards, as detailed below.

1. Describe how the normality of data distribution was assessed. 

2. If the signal from the same electrode/channel was tested under different conditions in the experimental setup, the authors should use repeated ANOVA to analyze the data, treating varied recording conditions as a within-subjects factor.

3. Key findings could be presented in graphical format.

Author Response

Please see PDF file attached

Reviewer 2 Report

In this manuscript “Value of mini electrodes for mapping myocardial arrhythmogenic substrate – Influence of tip-to-tissue angulation and irrigation flow on signal quality”, Karen Bickel et al. systematically investigated the effect of mini electrodes configuration (interconnection, angulations, contact forces, etc) and irrigation flow on the signal quality of electrograms. The results would certainly be of broad interest, as they are valuable and useful for clinical practice.

Apart from this decision, I have a few comments (below).

1. The major results of this manuscript are statistics, the conclusions would be more convinced if typical electrograms are provided to intuitively describe the observations.

2. The numerical data should be displayed in the form of curves or figures, rather than mostly tables as in the current manuscript.

3. As an animal test, not only the surgical procedure but also the chemicals involved in the experiment should be provided, for better reproducibility.

Author Response

Please see PDF file attached

This manuscript is a resubmission of an earlier submission. The following is a list of the peer review reports and author responses from that submission.

Round 1

Reviewer 1 Report

Bickel and colleagues report on a study investigating the impact of catheter contact, catheter angulation and irrigation on signal quality of conventional and mini-electrodes

They performed epicardial sampling of the left atrium on a pig model (single pig as I understand it) with 63 sampling points. 

Major comments:

1. With only 63 samples from a single pig it is difficult to make conclusions from this data

2. The authors took a large no. of samples from a relatively small area - it would be beneficial to indicate where on the atrium the samples were taken from (perhaps a figure?). Ultimately, of a large number of samples have bee taken from a small area, the incremental value of each sample is limited

3. This comment seems vague:

In a pre-test it was assured that there were no significant differences 88 between signals derived from mini electrodes from endocardium and epicardium. 

Where is the data to support the comment (particularly given the small sample size)

Minor comments:

1. It is important to indicate that this data is more relevant to the specific catheter design used in this study - other micro-electrode catheters (e.g. QDOT) have different designs and would have a different effect based on angulation and CF

2. This line gives the impression that you mounted the electrodes on a conventional catheter. Maybe preferable to say that a commercially available catheter with mini-electroges was used:

' The mini electrodes 21 were mounted on a commercially available 8 mm non-irrigated and 4 mm irrigated tip catheter '

Reviewer 2 Report

Major comments: The authors of the study analyzed the effect of catheter angulation and irrigation flow on the signal quality of the bipolar egms derived both from minielectrodes and conventional bipoles.

Even though the topic could be of interest, the findings of the study have very limited clinical utility for several reasons: 1) the tested catheters have no contact force sensors, so the contact of these catheters with the tissue was evaluated visually without any quantitative measurements and this could have had a high impact on  the findings of the study 2) The authors chose to use an irrigation flow that is lower to the recommended one for this catheters during ablation, so we have no information if the findings of the study still hold true in clinical practice.         In general, many statements in the paper were not quantitatively proved (see all sentences regarding robustness to contact force etc) and this is a significant limitation

Minor comments

Line 47- 48 : Authors should indicate the name of the catheter having the features they describe, because this is not true for every catheter with minielectrodes

Line 61: please correct english sentence wording (for instance , “shorter radiofrequency time and time … ”)

Line 87-88: The authors state that “ no significant differences  between signals derived from mini electrodes from endocardium and epicardium”,  but it is not clear which features of the signals were compared (amplitude, duration ? )so please expand this concept further or delete this sentence

Line 99: The author state that the metric pump used was  the Osypka. What is the reason the didn’t use the METRIC Pump from Boston Scientific? Could the use of this different pump could have an impact on the study results? Probably the answer is no, but this should be stated in the study limitations

Line 111- 114 : The authors mention contact force in those lines, but the catheters are not equipped with contact force sensors so it is not clear to me how the contact force was assessed during the experiments.

Line 128-129): The authors tested the irrigation flow of 10 ml/min that is not the recommended irrigation flow by Boston Scientific during ablation with this catheter. The authors should have tested a higher irrigation flow to simulate what happens with ablation with these catheters.

Line 244-245: The authors state that this is the first paper to investigate effects of contact force on signal morphology. Actually this statement is not supported by the findings of the paper for several reasons: the catheters tested have no contact force sensors and it is unclear how the authors measured the applied force to the epicardium, in addition to that, I don’t see any tables comparing signals features in different contact force settings so I couldn’t  understand what the authors mean in these lines

Lines 261- 264. These should not be part of the discussion session and should be moved to the limitations of the study.

Line 264- 267: The authors didn’t test signal quality during ablation and they didn’t use neither the recommended irrigation flow  pump for these catheters nor the irrigation flow suggested for ablation, so they can’t claim “Therefore one may  conclude that mini electrodes seem to be more appropriate for signal analysis during  ablation compared to conventional electrodes”. This claim has not been tested or validated in the study

Lines 284- 295: These concepts have nothing to do with the findings of the study and have already been quoted in the introduction session, so I suggest that the authors delete this part of the discussion

Line 324: every comment about the effect of contact force on signal quality,in every session of the paper (Abstract, discussion, conclusion etc) is not supported by the findings of the study as already explained in previous comments